# Study on Temperature-Dependent Uniaxial Tensile Tests and Constitutive Relationship of Modified Polyurethane Concrete

**DOI:** 10.3390/ma16072653

**Published:** 2023-03-27

**Authors:** Yanqun Han, Xiandong Meng, Fan Feng, Xuming Song, Fanglin Huang, Weibin Wen

**Affiliations:** 1School of Civil Engineering, Central South University, Changsha 410075, China; 2School of Architectural Engineering, Hunan Institute of Engineering, Xiangtan 411104, China; 3Hunan Tieyuan Civil Engineering Testing Co., Ltd., Changsha 410004, China

**Keywords:** constitutive relation, digital image correlation, modified polyurethane concrete, temperature, tensile mechanical properties, uniaxial tensile experiment

## Abstract

Modified polyurethane concrete (MPUC) is a new material for steel deck pavements. In service, the pavement is often cracked due to excessive tensile stress caused by temperature changes. In order to study the tensile properties of MPUC in the diurnal temperature range of steel decks, uniaxial tensile tests of MPUC were carried out at five temperatures. Three kinds of specimens and a novel fixture were designed and fabricated to compare the results of four different tensile test methods. The deformation of the specimen was collected synchronously by two methods: pasting strain gauge and digital image correlation (DIC) technique. Based on the experiment, the tensile mechanical properties, failure modes, and constitutive relations of MPUC were studied under the effect of temperature. The research results show that the novel fixture can avoid stress concentration. By observing the fracture surface of the specimens, the bonding performance is great between the binder and the aggregate at different temperatures. The tensile strength and elastic modulus of MPUC decrease with increasing temperatures, while the fracture strain, and fracture energy increase with increasing temperatures. The formulas of temperature-dependent tensile strength, fracture strain, and elastic modulus of MPUC were established, and the constitutive relationship of MPUC is further constructed in the rising stage under uniaxial tension. The calculation results show good agreement with experimental ones.

## 1. Introduction

As part of long-span steel bridges, steel bridge deck pavement (SBDP) is often paved on orthotropic steel bridge decks (OSBDs) to avoid damage to the OSBDs under vehicle load and harsh environments [1,2]. During the operation service period, the pavement materials should have sufficient strength, impact resistance, wear resistance, flexibility, and durability alongside good high-temperature stability and low-temperature crack resistance. Over the past half-century, gussasphalt (GA), stone mastic asphalt (SMA), and polymer-modified asphalt, which includes rubber, epoxy, and polyester, have been widely used in SBDPs [3,4]. The abovementioned materials used in bridge pavements have the disadvantages of having low strength and poor durability, and being temperature sensitive [5,6]. Therefore, further development of high-performance materials for SBDPs is essential for bridge safety and deck maintainability.

Polyurethane (PU), a polymer containing polyester diol and diisocyanate, and with a carbamate group, was synthesized by O. Bay in 1937 [7,8]. The main chain of the PU molecule is composed of hard and soft segments, which make PU have high mechanical strength and hardness. Meanwhile, it has excellent properties such as elasticity [6], low-temperature flexibility [9], good wear resistance [10], and chemical corrosion resistance [11]. Subsequently, researchers have developed polyurethane mixtures suitable for SBDPs, such as polyurethane mortar, polyurethane concrete (PUC), polyurethane ceramic recycled aggregate concrete [12], etc. PUC appeared earlier and was prepared by mixing traditional aggregates and polyurethane for the foundations, structural reinforcement, and pavement maintenance [13]. Owing to the excellent properties and the simple construction process of PUC, researchers have studied it in depth. Cong et al. [14] studied the tensile uniaxial properties of two kinds of porous polyurethane concrete at room temperature. It was found that modified PU could significantly change the tensile property of PUC. However, the increase in strength led to the PUC brittleness increasing. Further, Lu et al. [15] carried out a series of low-temperature fracture tensile and uniaxial compression tests on new polyurethane-bound pervious mixtures (PUPMs). The results showed that the tensile and compressive properties of the material were excellent, while PU and the aggregate were well bonded. The main reason for the failure of PUPM is an aggregate fracture. Jia et al. [16] studied the uniaxial tensile properties of polyurethane concrete doped with rubber particles at room temperature and proposed a linear elastic stress–strain relationship calculation formula. Pi et al. [17] carried out uniaxial tensile and compressive performance tests of polyurethane concrete doped with basalt, polyethylene fiber, and rubber powder. Based on Sargon’s classical mechanics constitutive model, uniaxial compression constitutive relations were proposed for three kinds of PUC. Owing to the complexity of the uniaxial tensile test, many researchers have also adopted indirect tensile test methods to learn the tensile mechanical properties of various polyurethane concretes, such as the bending test [18] and splitting tensile test [19], quickly and conveniently.

In summary, there is much research focusing on the mechanical properties and engineering applications of PUC materials at low and room temperatures. In engineering, pavement materials often soften at high temperatures and crack at low temperatures. A good pavement material should exhibit good mechanical properties, stability, and crack resistance within the temperature range of service. Currently, there are many projects that use PUC materials. However, there is limited research on the high-temperature tensile property and temperature-dependent constitutive relationship of PUC. Therefore, it is significant to provide a theoretical basis for the mechanical analysis and life prediction for the PUC pavement layer.

This paper is organized as follows. In Section 2, the test materials and methods used in the study are described. Three specimens and a new type of fixture were designed and manufactured to compare the results of four different tensile test methods. Tensile tests were conducted at various temperatures using the optimal method, and the mechanical properties of the material were obtained. In Section 3, the formulas for calculating the temperature-dependent tensile strength, fracture strain, and modulus of elasticity of MPUC were established. In Section 4, a temperature-dependent tensile constitutive model for MPUC was established and compared to the constitutive model in the Chinese Code. Section 5 provides conclusions and prospects for future research.

## 2. Materials and Methods

### 2.1. Material and Proportioning

The main components of MPUC include aggregates, MPU, and curing agents. According to the maximum density curve, the aggregate is mixed with coarse aggregate and fine aggregate, and the moisture content was less than 0.3%. The coarse aggregate is natural gravel with continuous grading, the particle size range was ~4.75–9.5 mm. The particle surface of coarse aggregate was rounded and smooth, and the content of needle-like particles was less than 8%, as shown in Figure 1a.

The fine aggregate is natural river sand with continuous grading, and the particle size is less than 4.75 mm, as shown in Figure 1b. The grading curves of aggregates are shown in Figure 2. The sieve pass rate of each grade curve is shown in Table 1.

MPU is a thermosetting polymer, which is a polyurethane material modified with a silane coupling agent (SCA). Polyurethanes are prepared from a variety of polymers, such as isocyanates, polyether-type diols, tertiary alcohols, stabilizers, etc. Then, MPU was obtained by adding the silane coupling agent. SCA, molecular formula Y-Si (OR)_3_, contains both inorganic and organic functional groups and is compatible with inorganic and organic materials [20,21]. After mixing MPU with an aggregate, SCA can form a strong covalent bond between PU and inorganic aggregate, which enhanced the interfacial strength between PU and the aggregate [22,23]. The reaction mechanism of SCA is shown in Figure 3.

The curing agent, which makes MPUC cure faster, is composed of toluene diisocyanate, hexamethylene diisocyanate, and other materials mixed in proportion. MPUC reaches 75% of its strength after 2 h of maintenance at room temperature [24]. The slice of the specimen is shown in Figure 1c. The materials for the test were provided by the Roaby Company (Ningbo, China). The ratio of MPUC is provided by Roaby, which applied the material to the pavement engineering, as shown in Table 2.

### 2.2. Tensile Specimen and Fixture Design

Compared with compression, there are some difficulties in the uniaxial tensile test of concrete. For example, whether the specimen fractures within a reasonable range is related to its shape. Dumbbell [25], prism [26], and thin plate specimens [27] are conducive to obtaining better test results and increasing the success rate of the tensile test in existing studies. However, the dimensions of tensile specimens are generally small and size effects are prevalent. Therefore, designing a reasonable specimen size and shape is the key to the success of the tensile test. The connection between the fixture and the specimen is also the key to the PUC uniaxial tensile test. The tensile test methods can be divided into fixed, top glued, and anchored, depending on the connection of the specimen and fixture [28]. The fixed method is often applied to embed the reinforcement at both ends of the specimen during fabrication. When the tensile test is conducted, the specimen is damaged by pulling the reinforcement through the testing machine. The top-glued method uses high-strength adhesive to bond the specimen with the fixture. However, this method is not suitable when the tensile strength of the specimen is greater than the bond strength of the high-strength adhesive. The anchored method is to anchor or clamp the specimen of the fixture to the machine. However, the contact surface between the fixture and the specimen often had a stress concentration during the test, which causes the specimen to fail before reaching the ultimate bearing capacity. In addition, the test researchers are heavily burdened by the weight and size of the tensile fixture; therefore, this paper seeks to find a lightweight and efficient tensile fixture for their study. Experiments were conducted to analyze the specimen shape and fixture forms. Both MPUC and ordinary concrete are non-homogeneous materials. For tensile specimens made of such materials, the width of the test area should be greater than 3 times the size of the largest aggregate and the length should be ~3–10 times the size of the largest aggregate [29,30]. In this study, three types of tensile specimens were designed and fabricated. Specimen 1 (S1) is a cylindrical specimen with a diameter of 150 mm and a height of 300 mm. Specimens 2 and 3 (S2 and S3) were dumbbell-type specimens. The two ends of S2 and the cross-sectional dimensions of the tensile test area were 150 mm × 75 mm and 76 mm × 75 mm, respectively, with a straight transition from the two ends to the test area. The cross-section size of both ends of S3 was the same as that of S2, and the tensile test area was 60 mm × 75 mm, with a combination transition of a straight line and arc from both ends to the test area. The shapes of the specimens are shown in Figure 4.

The specimen length is the key to the effect on the test results. If the specimen length is too short, the end effect is obvious, and a fracture often occurs at the end of the specimen. When the specimen length is too long, the self-weight of the specimen is too large, which is not conducive to installation and testing [31]. Empirically, the test area lengths of S2 and S3 were designed to be 120 mm and 60 mm, respectively, while the lengths of both specimens were 300 mm, as shown in Figure 4. In order to realize the tensile of the specimens, three tensile test fixtures (I, II, and III) were designed and fabricated in this study. Type I fixture was used for cylindrical specimens. Types II and III fixtures were used for dumbbell-type specimens. Type II had no steel bolt and lumbar foramen compared with type III. The fixture diagram is shown in Figure 5 and Figure 6.

The following four MPUC tensile test schemes were obtained by combining the designed specimens and fixtures, as shown in Table 3. In Table 3, the symbols are explained in the Appendix A. The same below.

Tensile testing was carried out with reference to the GB/T 50081-2019 (China’s Ministry of Housing and Urban-Rural Construction, 2019), and displacement loading was used for all kinds of tensile tests, with a loading speed of 1 mm/min.

As can be seen from Table 3, adhesive strength is the key to the success of SJ-1. According to the test results of SJ-1, the bonding strength of epoxy resin adhesive was greatly affected by the test temperature of this paper, and the bonding failure often occurred at high- and low-temperature states. For SJ-2–SJ-4, the difficulty is how to avoid the stress concentration at the interface of the specimen and fixture. The test area of SJ-2 transits to both ends in a straight line, which leads to the formation of an angle between the transition area and the test area, resulting in stress concentration. During the SJ-2 tensile process, the specimen and the steel fixture became closer and closer, which led to the specimen cracking at the corner point, meaning the tensile strength did not reach the ultimate strength at the time. Compared with SJ-2, SJ-3 changes the straight-line transition to a straight line with an arc, which can effectively relieve the stress concentration, as shown in Figure 7a. To verify this problem, two finite element models of SJ-2 and SJ-3 were established. The calculation results show that the area within the two blue dotted lines of SJ-2 has high damage, as shown in Figure 7a. This result indicates that SJ-3 can better avoid stress concentration.

The main crack of SJ-3 is within the gauge, as shown in Table 3. However, as shown by the test results of SJ-2 and SJ-3, arc-shaped cracks exist at both ends of the specimen. During the test, the fixture was deformed and the specimen rotated along the loading axis. The contact mode changed from the surface to the line (point in Figure 7b) between the specimen and fixture, which caused the stress concentration and the specimen end to crack, as shown in Figure 7b.

According to the problems in SJ-2 and SJ-3, every fixture should add two screw bolts (10 mm in diameter) and four screw nuts to reduce the deformation, which improved the restraint effect of the fixture on the specimen. According to the results in Table 3, the main crack in the specimen is within the measurement length, while no cracks appeared at the end in SJ-4. Since the screw bolts and screw nuts increase the contact force between the specimen and the fixture, the stress concentration did not occur at the end of the specimen.

### 2.3. Temperature Dependent Tensile Test

#### 2.3.1. Specimen Production

Mix screened and weighed aggregates, modified polyurethane binder, and hardener were put in a high-speed concrete mixer for 3–5 min. Based on the design of S3, the MPUC mixture was loaded into a steel mold and vibrated on a shaking table. In this experiment, the ambient temperature was about 14 to 17 °C. The molds were removed after 24 h and all specimens were tested after 3 days.

#### 2.3.2. Test Procedure

Based on the scheme of SJ-4, the uniaxial tensile test of MPUC was carried out at different temperatures, with the test procedure shown in Figure 8.

Firstly, the size and quality of the specimens were reviewed, and the specimens with large deviations were removed. The calculated density of MPUC ranged from 2150 to 2250 kg/m^3^, with an average density of about 2200 kg/m^3^. Secondly, the tensile strains of the specimens were measured by both a non-contact video extensometer (DIC) and resistive strain gauges. Preparations for DIC technology: 1. Spray white paint on the specimen surface in advance; 2. Randomly spray black spots of certain deformation capacities; 3. Set up a high-frequency camera. The black dots on the specimen surface were identified and the relative displacements between the black dots were calculated by digital image processing techniques during the tensile process. After further calculation, the displacement field, deformation gradient, and strain field of the specimen were obtained in a specific time interval. The non-contact measurement system, VIC-3D manufactured by Correlated Solutions, Inc., was used to identify the black dots and calculate them. Three foil strain gauges were pasted on the back of the specimen. The size of the strain gauge sensitive gate was 50 mm × 4 mm, the resistance was 120 Ω, and the sensitivity was 2.0 ± 1%. The strain data, measured by the strain gauge, were used to check the data of the DIC. The arrangement position of the strain gauge and DIC scatter point is shown in Figure 8.

During the operation, the temperature range of the steel bridge deck pavement was −10 °C to 70 °C [32]. MPUC is a new pavement material, thus, in order to study the uniaxial tensile properties of MPUC with temperature, the tests were carried out at five temperatures: −10 °C, 0 °C, 15 °C, 40 °C, and 60 °C. Each group of temperature tests prepared 3 specimens, with a total of 15 specimens. The specimens were left to stand for 6 h at the test temperature, to ensure the same temperatures inside and outside of the specimen [33]. In order to steadily raise (fall) the temperature of the specimen, the temperature rise (fall) rate of the thermotank (JK-HW-150L) was 1 °C/min. The laboratory temperature was about 15 °C. Figure 9 shows the temperature change process in the thermostat.

The humidity was not set between −10 °C and 0 °C, while the relative humidity was set to 70% (ambient humidity) at the other three temperatures. Before the test, a high-frequency camera was established for the DIC and image calibration. The specimen was removed from the thermostat and the strain gauge was connected to a strain-gathering instrument (HBM MGC plus), which was set to a sampling frequency of 20 Hz. The test was carried out using a 1000 kN microcomputer-controlled electro-hydraulic servo testing machine (WAW-1000-G) for loading. All specimens were loaded with a displacement control load at a rate of 1 mm/s. The specimen was pretensioned during loading. The threshold value of the pretension load was 5 kN, to ensure tight contact between the loading surface and the fixture surface before formal loading [34]. Finally, the tester continued to load uniformly until the specimen was damaged, and the load value and fracture position of the specimen were recorded at the time of damage.

#### 2.3.3. Test Phenomenon

Through testing, it was found that the higher the temperature was, the greater the deformation of the specimen became during the tensile process. After the damage of the specimens, the main cracks in most of the specimens were distributed within the gauge, while the fracture surface was flat, which showed that the specimens were uniformly stressed in the tensile axial direction. Very few specimens showed arc-shaped cracks at the end due to stress concentrations caused by manufacturing errors. There were no obvious microcracks near the fracture interface (within the gauge). The specimen was subjected to axial force without bias tension, which was typical for uniaxial tensile damage. The fracture cross-section showed that the fracture occurred mostly at the adhesion between the aggregate and MPU, with a small number of holes produced by the fine aggregate shedding and a few coarse aggregates being pulled off. A schematic diagram is shown in Figure 10.

As the temperature increased, the amount of the damaged coarse aggregate did not decrease, nor were the number of holes significantly increased. The increase in the temperature did not affect the bonding effect between the MPU and the aggregate. Only very few bubble holes were found on the fracture surfaces of the individual specimens, which indicates good compactness in MPUC. The fracture cross-section of the specimen is shown in Figure 11.

The strain field of the specimen was calculated by the data of DIC, and the fracture position of the specimen was compared with the test results. The strain field of the specimen matched well with the test results, as shown in Figure 12.

#### 2.3.4. Test Data Results

The test results of the tensile peak stress, peak strain, elastic modulus, and tensile fracture energy at various temperatures are shown in Table 4. The tensile performance indicators at each temperature are the average values measured for the test specimens in that group.

According to the test results, it was found that the tensile performance indexes of MPUC were significantly affected by the temperature. Each temperature in Table 4 corresponds to the average value of the tensile performance index of a group of specimens.

The peak strain values obtained by the two strain acquisition methods, strain gauge, and DIC, are provided in Table 4. The relative errors of the peak strains for both are less than 10%, which shows that the test results were good. However, compared to high temperatures, the relative error of the two strain measurements is larger at lower temperatures. The DIC value is smaller than the stain gauge value. The reason is that the tiny microcracks appeared at the ends of the specimen during loading. The tiny displacement caused by microcracks makes the DIC value smaller.

## 3. Analysis of Test Results

### 3.1. Stress-Strain Curve

Figure 13 shows the MPUC stress-strain curves at different temperatures. When the tensile specimen reaches the peak strain, the curves at the different temperatures decrease rapidly, and there is no traditional decline curve.

This paper mainly studied the effect of temperature on the rising section of the curve MPUC tensile stress-strain, thus, the peak strain is approximated as the fracture strain. As can be seen from Figure 13, the temperature had a large effect on the curve shape of the MPUC stress–strain. The slopes of the curves for −10 °C and 0 °C are similar. The stress and strain increase linearly in proportion to each other until the peak strain is reached. As the temperature increases, the initial slope of the stress-strain curve becomes smaller, the linear elastic section of the curve becomes shorter, the strain growth rate increases and the curve bends more obviously toward the strain axis. It shows that the higher the temperature softening effect is, the earlier the material enters the softening stage.

The tensile fracture energy density of the material can be calculated by the area surrounded by the stress-strain curve and the strain axis, as shown in Table 4. The fracture energy density can reflect the toughness characteristics of the material. The greater the fracture energy density is, the stronger the toughness is. As shown in Figure 13 and Table 4, the higher the temperature is, the higher the fracture energy density is. It shows that the more energy required for tensile damage per unit volume of MPUC, the stronger the toughness is. The reason for the above phenomena is that the increase in temperature helps the MPU molecular skeleton to break through the internal rotation barrier and enhance molecular movement. The work performed by the external load is converted into molecular kinetic energy and friction heat energy. Therefore, more energy is needed to break the specimen at high temperatures [35,36].

### 3.2. Tensile Peak Stress

Based on the test results, a relationship was obtained between the tensile peak stress of MPUC, and the test temperature values, as shown in Figure 14.

The results show that the tensile peak stress (tensile strength) of MPUC decreases with increasing temperature. The peak stress of MPUC at 60 °C decreases by 61.6% compared to −10 °C. MPU is an amorphous polymer whose intermolecular forces decrease with increasing temperature, which leads to the decrease of material strength from the macroscopic view [36]. When a load is applied, MPU cracks and becomes damaged before the aggregate, which affects the material tensile strength of MPUC.

Since MPUC is in a “glassy state” at −10 °C and 0 °C, the continuous drop in temperature has little impact on the mechanical property of MPUC. Therefore, the mechanical properties of MPUC are similar at these two temperatures. However, the tensile peak stress of MPUC at −10 °C is slightly less than that at 0 °C, which is the test error due to the discreteness of the specimen, as shown in Figure 14. The test data are fitted by the least squares criterion, where the variation of tensile peak stress with temperature is obtained as
(1)$$\frac{f_{t,T}}{f_{{t,T}_{0}}}=0.020{\left(\frac{T}{T_{0}}\right)}^{3}-0.125{\left(\frac{T}{T_{0}}\right)}^{2}+1.120$$
where *T* and *T*_0_ are the test temperature and 15 °C, respectively, and *T* takes a value range of −10 °C ≤ *T* ≤ 60 °C. The room temperature during the test was 15 °C. *f*_t,T_ and $f_{{t,T}_{0}}$ are the tensile peak stresses of MPUC at temperatures *T* and *T*_0_, respectively. The correlation coefficient is *R*^2^ = 0.999 and the residual sum of squares is *v*^2^ = 2.145 × 10^−5^, between the fitted curve and the test data. The peak stress fitting agrees well with the test results.

### 3.3. Tensile Peak Strain

The variation of MPUC tensile peak strain at each temperature is provided in Figure 15.

The peak strain is the arithmetic mean of the strain gauge data and DIC values. As can be seen from Figure 15, the peak strain of MPUC increases with increasing temperatures. The higher the test temperature is, the more obvious the increase in peak strain is. The peak strain at 40 °C is 127.6% higher than at −10 °C, while the peak strain at 60 °C is 232.9% higher than at 40 °C. This phenomenon is related to the temperature characteristics of MPU in MPUC. The empirical equation for the tensile peak strain of MPUC with temperature is established using a least squares criterion of:(2)$$\frac{\varepsilon_{t,T}}{\varepsilon_{{t,T}_{0}}}=0.030e^{1.260\frac{T}{T_{0}}}+0.690$$
where *ε*_t,T_ and $\varepsilon_{{t,T}_{0}}$ are the tensile peak strains of MPUC at temperatures *T* and *T*_0_, respectively. The correlation coefficient is *R*^2^ = 0.998 and the residual sum of squares is *v*^2^ = 2.370 × 10^−4^ between the fitted curve and the test data. The peak strain fitting agrees well with the test results.

### 3.4. Elastic Modulus

The tensile elastic modulus of MPUC at each temperature is provided in Figure 16.

The elastic modulus of MPUC decreases as the temperature increases, and the elastic modulus of MPUC decreased by 92.8% at 60 °C compared to −10 °C. The temperature change causes a change in the degree of molecular motion for MPU. Thus, at low temperatures, the energy of the molecular thermal motion is low, and the molecular struggles to overcome the internal rotation barriers in the backbone. When MPU is subjected to an external force, the internal rotation of the single bond of the molecular chain makes the chain segment move. It is the change in ‘conformation’ that adapts to the action of the external force. Part of the energy of the external force work is converted into heat energy due to the internal friction between the molecules. The increase in the internal energy of the material is accompanied by a decrease in the intermolecular force. Macroscopically, the decrease in strength and elastic modulus lead the deformation to increase and the material properties to change from elastic brittleness to high elasticity [35,36].

By analyzing and fitting the test data, the calculation formula of the elastic modulus of MPUC changes with temperature is obtained, as shown in Equation (3). The fitting curve is compared with the test data, as shown in Figure 16.
(3)$$\frac{E_{t,T}}{E_{{t,T}_{0}}}=-0.290\frac{T}{T_{0}}+1.290$$
where *E*_t,T_, *E*_t,T_0__ are the elastic modulus of MPUC at temperatures *T* and *T*_0_, respectively. The correlation coefficient is *R*^2^ = 0.992, and the sum of residual squares is *v*^2^ = 0.010. The elastic modulus fitting is in good agreement with the test results.

## 4. Constitutive Relationship

### 4.1. Uniaxial Tensile Constitutive Model

At present, MPUC has a broad application prospect in steel bridge deck pavements. Studying the tensile constitutive model of MPUC under temperatures is helpful to evaluate the status of the pavement deck. It can guide the numerical simulation of bridge deck structures and the prevention of pavement damage. The fiber bundle model based on damage theory is often used to describe the tensile damage process of composite materials, such as concrete. The model diagram is shown in Figure 17.

The model simplifies the material to include ***M*** independent fiber elements. The elements are randomly damaged, and the stress-strain curve of the damaged elements is linear. When ***m*** elements are damaged, the external load will be uniformly distributed over the remaining elements (***M*** − ***m***). The damage variable (***D***) is usually defined as the ratio of the number of damaged elements (***m***) to the total number of elements (***M***). The stress-strain relationship under arbitrary loading conditions is shown in Equation (4).
(4)σ=Eε(1−D)

To accommodate the constitutive relation of the different types of concrete under different test conditions, the value functions of the damage variables are different. The damage principal structure model for ordinary concrete in the Chinese Code for the Design of Concrete Structures (GB 50010-2010)-2015 has damage variables as shown in Equation (5). However, the effect of temperature is not considered in the definition process of damage variables in the code [37].
(5)$$D=1-\frac{\sigma_{t}}{E\varepsilon_{t}}(1.2-0.2{\left(\frac{\varepsilon }{\varepsilon_{t}}\right)}^{5})$$
where *σ* and *ε* are the tensile stress and strain of concrete, respectively; *σ*_t_ and *ε*_t_ are the tensile peak stress and peak strain of concrete, respectively; *E* is the modulus of elasticity of concrete.

In this study, based on the fiber bundle model and experimental data, the temperature parameters are proposed as *α* and *β*, and the temperature-dependent damage variable is provided. The intrinsic relationship of MPUC under the effect of temperature is shown in Equations (6) and (7).
(6)$$\sigma =E_{t,T}\varepsilon (1-D_{t,T})$$
(7)$$D_{t,T}=1-\frac{f_{t,T}}{E_{t,T}\varepsilon_{t,T}(\alpha {\left(1-\frac{\varepsilon }{\varepsilon_{t,T}}\right)}^{\beta }+\frac{\varepsilon }{\varepsilon_{t,T}})}$$
where *D*_t,T_ is the damage variable of the temperature correction. After substituting Equation (7) into Equation (6), Equation (8) is obtained. In Equation (8), *f*_t,T_, and *ε*_t,T_ are calculated by Equations (1) and (2), although *α* and *β* are unknown. When *ε* = 0, *dσ*/*dε* = *E*_t,T_. Therefore, *α* could be obtained by using Equation (9).
(8)$$\sigma =\frac{f_{t,T}\varepsilon }{\varepsilon_{t,T}(\alpha {\left(1-\frac{\varepsilon }{\varepsilon_{t,T}}\right)}^{\beta }+\frac{\varepsilon }{\varepsilon_{t,T}})}$$
(9)$${\frac{d\sigma }{d\varepsilon }|}_{x=0}=\frac{f_{t,T}}{\alpha \varepsilon_{t,T}}=E_{t,T};\;\alpha =\frac{f_{t,T}}{E_{t,T}\varepsilon_{t,T}}=\frac{E_{\mathrm{tc},T}}{E_{t,T}}$$
where *E*_t,T_, and *E*_tc,T_ are the initial elastic modulus and peak secant modulus of MPUC at different temperatures, respectively, which are calculated by Equations (1)–(3). By bringing Equation (9) into Equation (8), the coefficient *β* could be obtained by Equation (8). The fitting results are shown in Table 5.

In the range of −10 °C to 15 °C, the coefficient *β* is almost unchanged following the increase in temperature. Therefore, *β* can be regarded as a constant, which is equal to the average of the three sets of temperatures, approximately *β* = 1. As the temperature continues to rise, the value of *β* changes significantly. The expression of the coefficient *β* is proposed:(10)$$\beta =\{\begin{array}{l} 0.03\frac{T}{T_{1}}+0.55\;15\;{}^{\circ}C<T\leq 60\;{}^{\circ}C\\ 1\;\;-10\;{}^{\circ}C\leq T<15\;{}^{\circ}C \end{array}$$
where *T* and *T*_1_ are the test temperature and 1 °C, respectively.

### 4.2. Comparison of Models

The constitutive model of concrete in the Chinese code is widely used in Chinese engineering. Figure 18 shows the calculation flow of the uniaxial tensile stress-strain constitutive model in the Chinese code and this study, respectively.

The peak stress, peak strain, and elastic modulus of the two models are the same at different temperatures, which are calculated using Equations (1)–(3) in this paper. In addition to the different constitutive equations, the constitutive model in the code does also not use temperature parameters.

The comparison between the MPUC constitutive model curves and the test curves is shown in Figure 19.

At low temperatures (−10 °C and 0 °C), the initial modulus of elasticity in the constitutive model curve in the code is larger than the experimental result, while the results are just the opposite at high temperatures (40 °C and 60 °C). The temperature-dependent constitutive relation is proposed in this study, where the calculation result is slightly different from the test curve in the softening stage at 40 °C and 60 °C. In general, the novel constitutive model better predicts the trend in strain development with stress in MPUC specimens under uniaxial tension. The calculated results of elastic modulus, peak strain, and peak stress are in good agreement with the experimental results. Compared with the model for the Chinese code, the MPUC tensile constitutive model proposed in this paper has a better prediction effect.

## 5. Conclusions

In this study, the effects of temperature on the uniaxial tension performance of the MPUC are investigated. The following conclusions can be drawn from the experimental data and analysis:A novel tensile test fixture (SJ-4) is developed. The fixture with the dumbbell-shaped specimens (arc transition) can effectively avoid stress concentration and ensure the specimen breaks within measurement length, which is suitable for the stretching of high-strength brittle materials.The bonding effect between MPU and aggregate is less affected by temperature. The tensile strength and elastic modulus of MPUC decrease with increasing temperature, while the fracture strain and fracture energy are the opposite. The variation in tensile strength, fracture strain, and elastic modulus of MPUC with temperature is well reflected by the proposed temperature-dependent equations, and the calculation results show good agreement with the experimental ones.The shapes of the tensile stress-strain curve of MPUC at low temperatures (−10 °C and 0 °C) and ambient temperature (15 °C) are similar to an elastomeric brittle material. As the temperature increases, the plasticity of MPUC increases. The relative error in measuring deformation with the DIC technique and strain gauges is related to the material properties. The stronger the plasticity of the test material is, the smaller the relative error is.The temperature-dependent uniaxial tension constitutive relation of the MPUC ascending segment is established. The prediction of MPUC is improved by introducing temperature-related parameters *a* and *b*, which are significantly better than the constitutive model for the Chinese code. The results provide a reference for the engineering application and numerical analysis of MPUC.

The key challenges of this study are the design of the fixture, the control of the specimen temperature during the experiments, and the establishment of constitutive relationships. During tensile testing, attention should be paid to the contact between the specimen and the fixture to avoid stress concentration.

## Figures and Tables

**Figure 1 materials-16-02653-f001:**
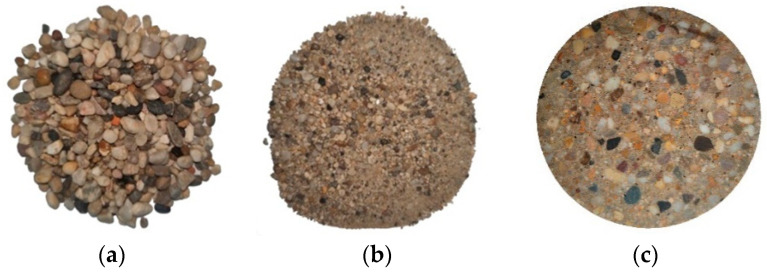
Aggregate and specimen slices, (**a**) coarse aggregate, (**b**) fine aggregate, (**c**) specimen section.

**Figure 2 materials-16-02653-f002:**
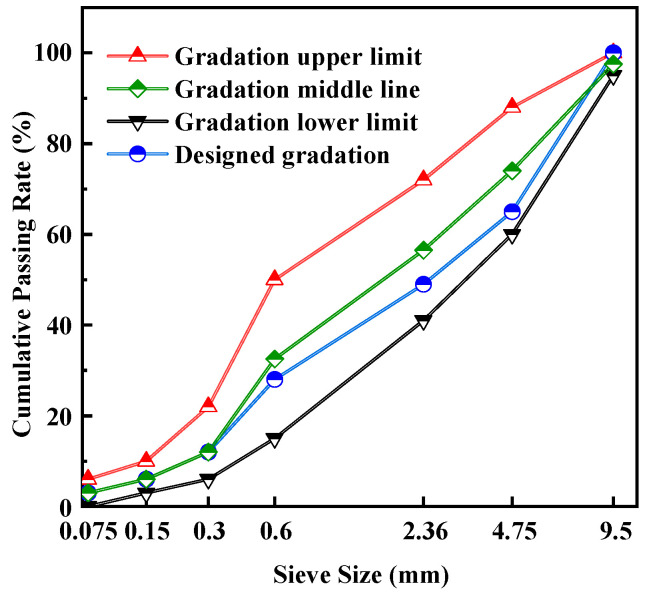
Gradation curves of mixtures.

**Figure 3 materials-16-02653-f003:**
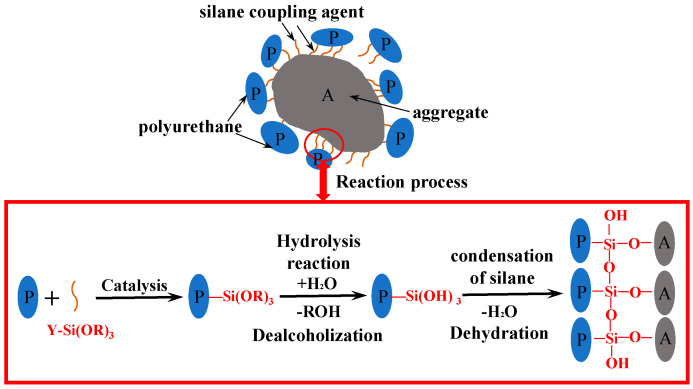
Reaction process of silane coupling agent.

**Figure 4 materials-16-02653-f004:**
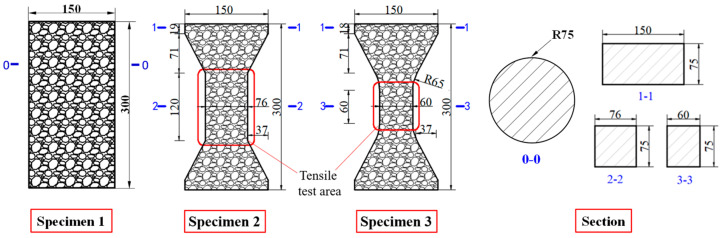
Shapes and sizes of specimens (unit: mm).

**Figure 5 materials-16-02653-f005:**
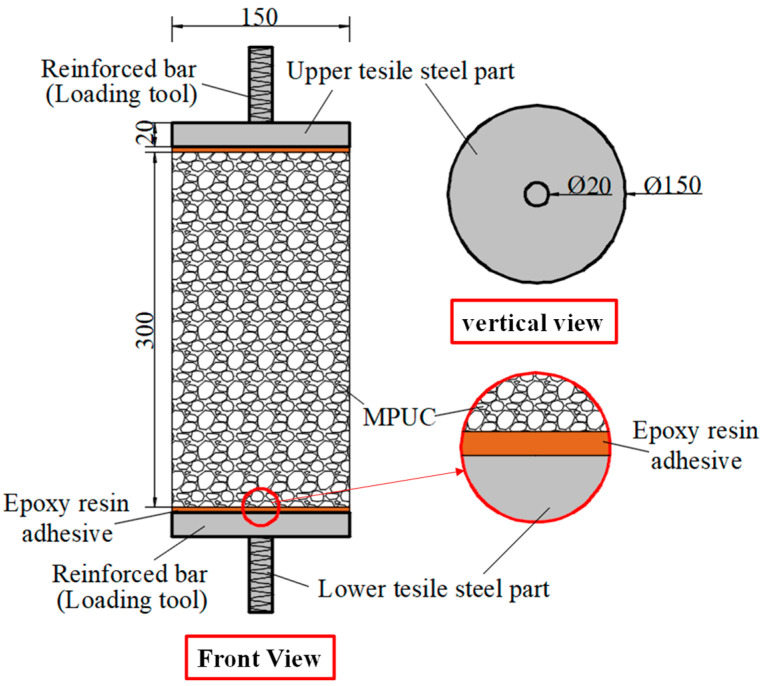
Direct tensile test fixture I (unit: mm).

**Figure 6 materials-16-02653-f006:**
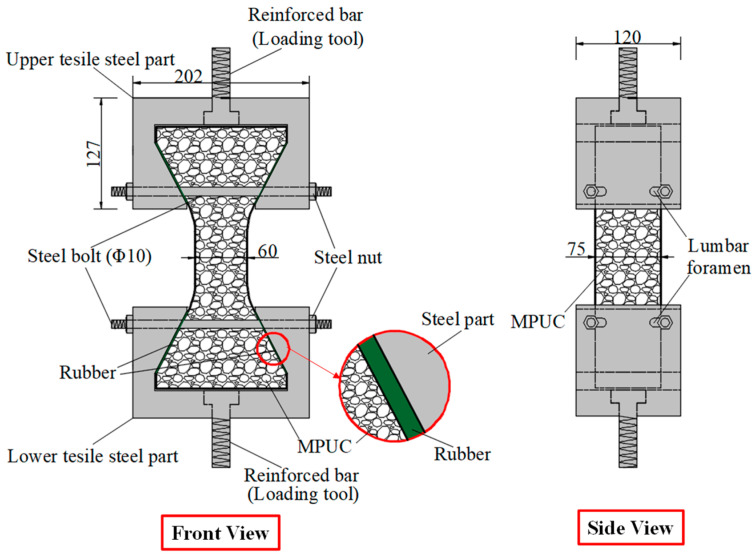
Direct tensile test fixture III (unit: mm).

**Figure 7 materials-16-02653-f007:**
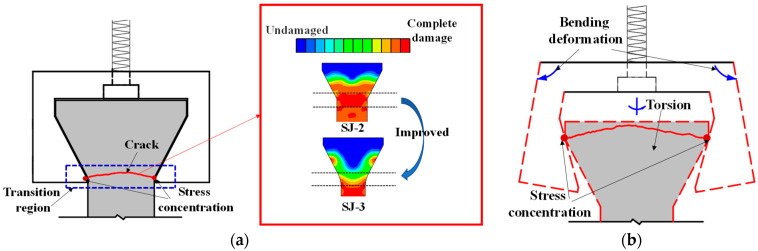
Schematic diagram of strain concentration of specimens, (**a**) transition zone fracture, (**b**) loading end fracture.

**Figure 8 materials-16-02653-f008:**
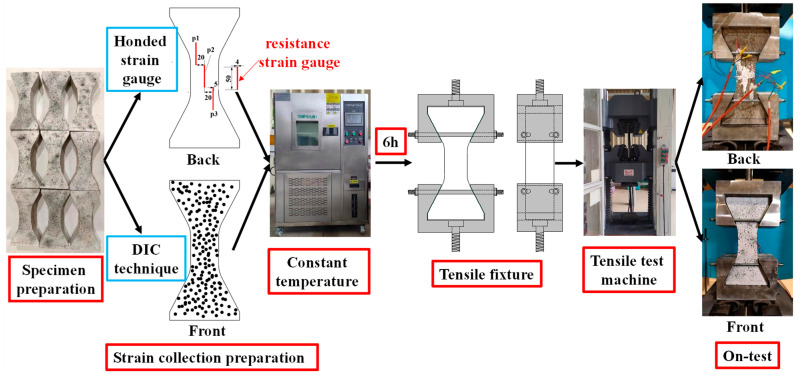
Experimental flowcharts.

**Figure 9 materials-16-02653-f009:**
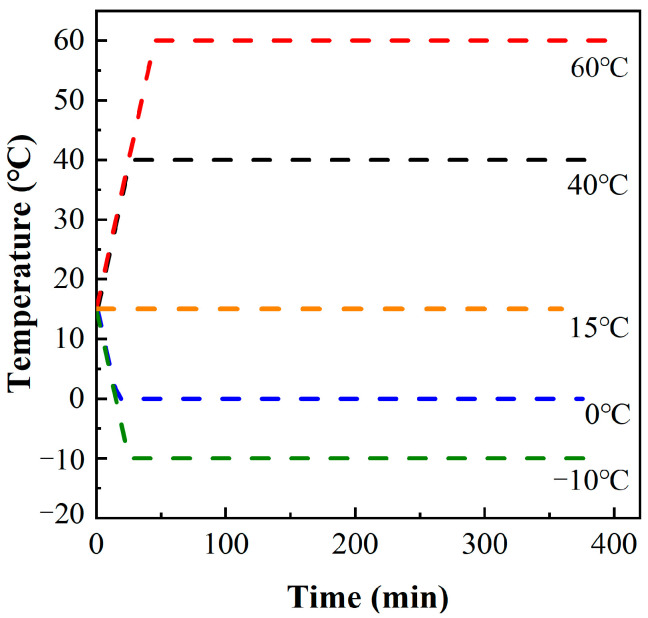
Temperature change curve of the furnace at the thermostat.

**Figure 10 materials-16-02653-f010:**
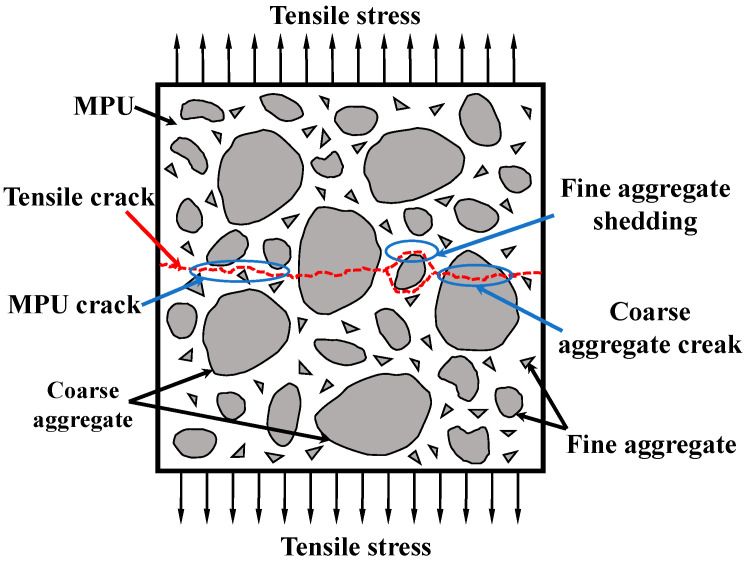
Schematic diagram of fracture section of specimen.

**Figure 11 materials-16-02653-f011:**
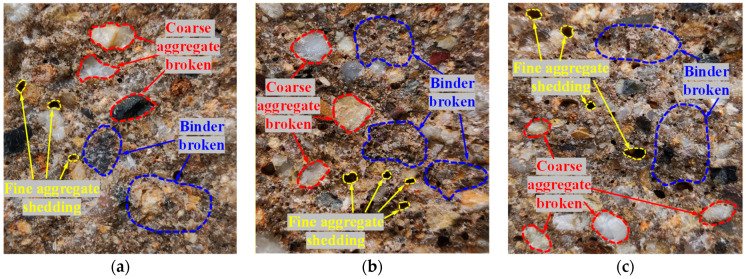
Fracture cross-section of the experimental specimen. (**a**) −10 °C, (**b**) 15 °C, (**c**) 60 °C.

**Figure 12 materials-16-02653-f012:**
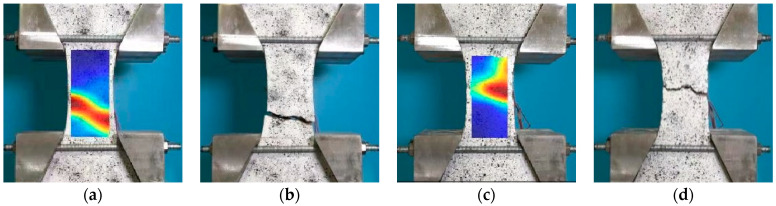
Comparison of specimen fracture and DIC digital imaging. (**a**) 40 °C—0.1 s before crack, (**b**) 40 °C—crack, (**c**) 60 °C—0.1 s before crack, (**d**) 60 °C—crack.

**Figure 13 materials-16-02653-f013:**
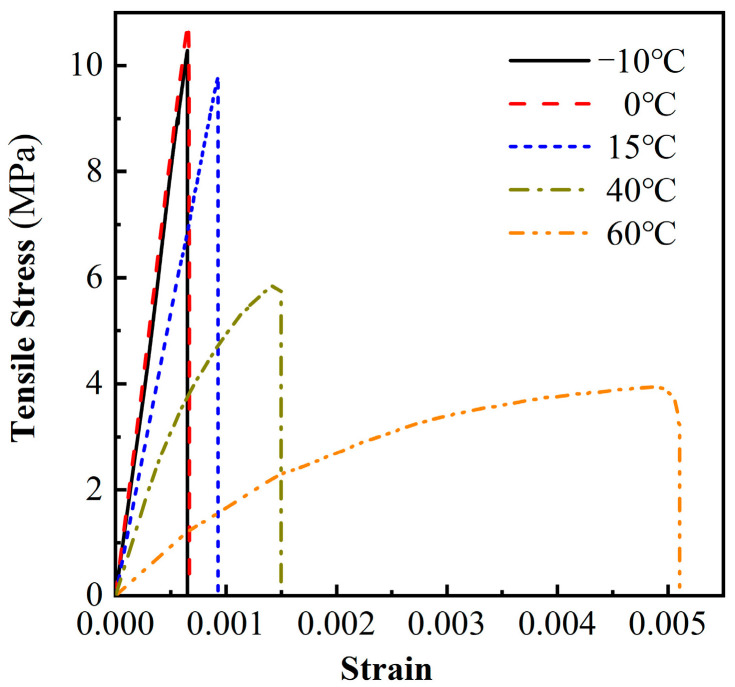
Stress-strain curves of MPUC at different temperatures.

**Figure 14 materials-16-02653-f014:**
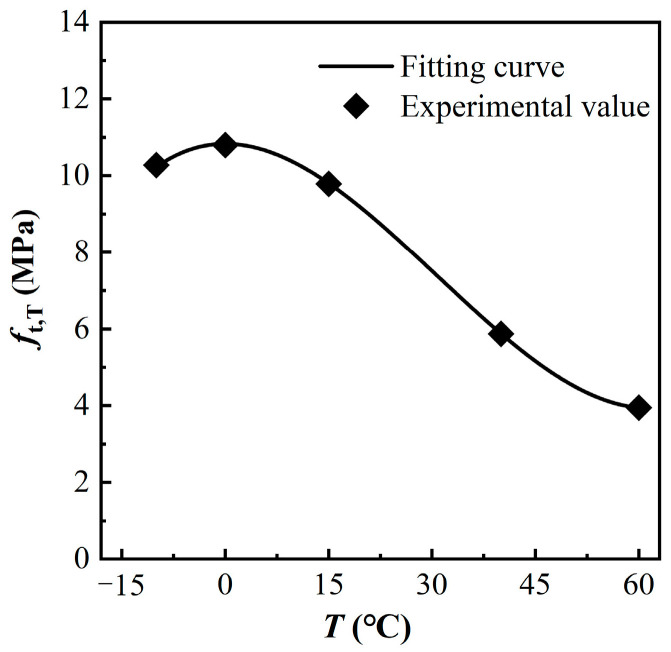
Relationship between tensile peak stress and temperature.

**Figure 15 materials-16-02653-f015:**
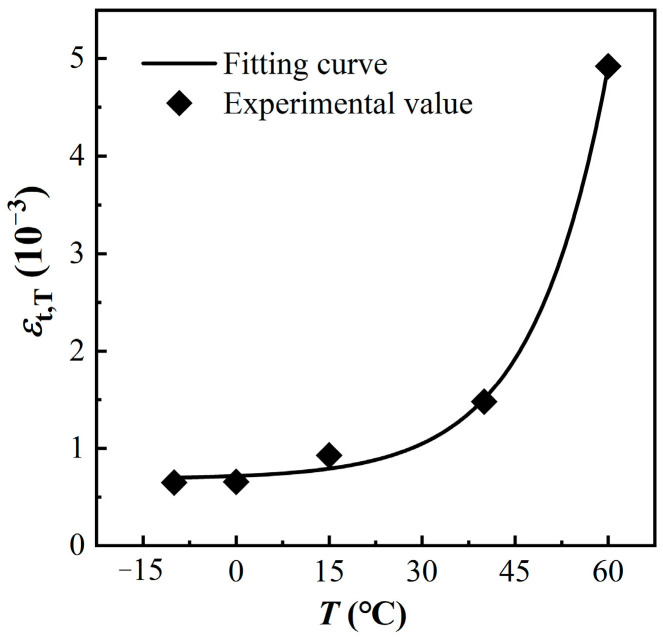
Relationship between tensile peak strain and temperature.

**Figure 16 materials-16-02653-f016:**
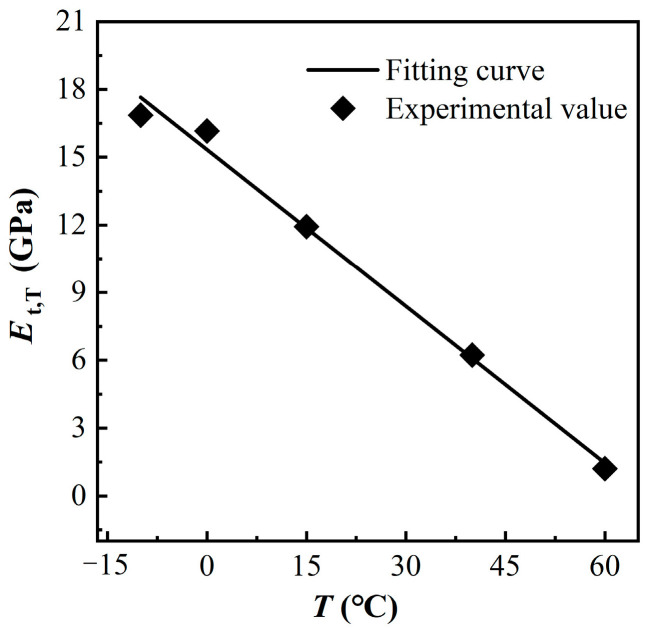
Relationship between tensile elastic modulus and temperature.

**Figure 17 materials-16-02653-f017:**
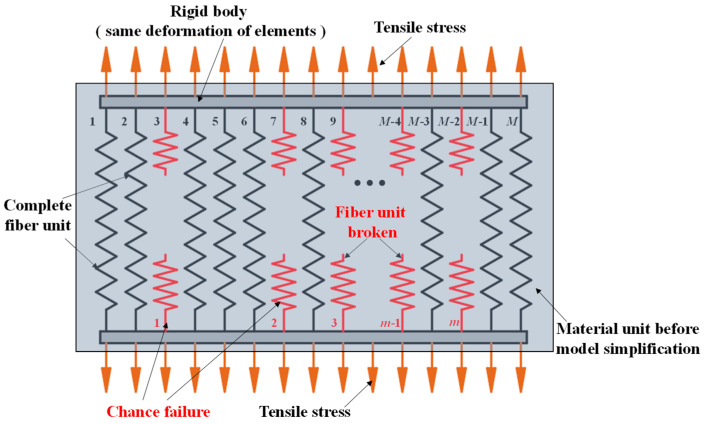
Tensile fiber bundle model.

**Figure 18 materials-16-02653-f018:**
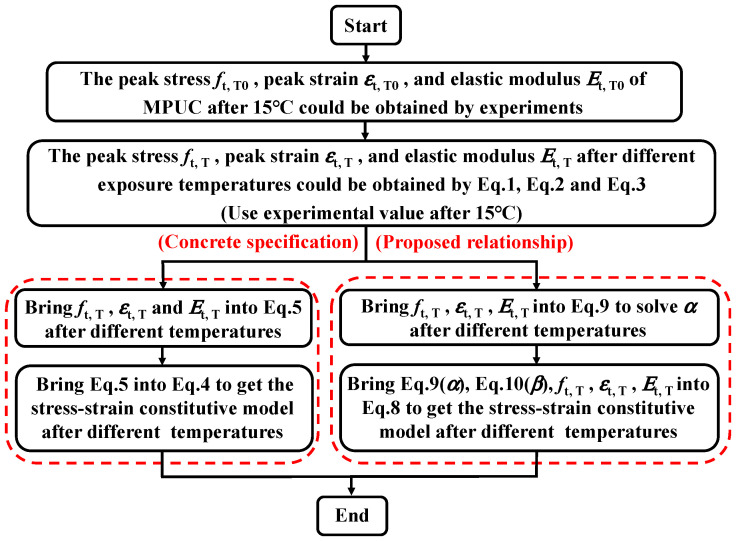
Comparison of algorithms for predicting stress-strain curves after different temperatures.

**Figure 19 materials-16-02653-f019:**
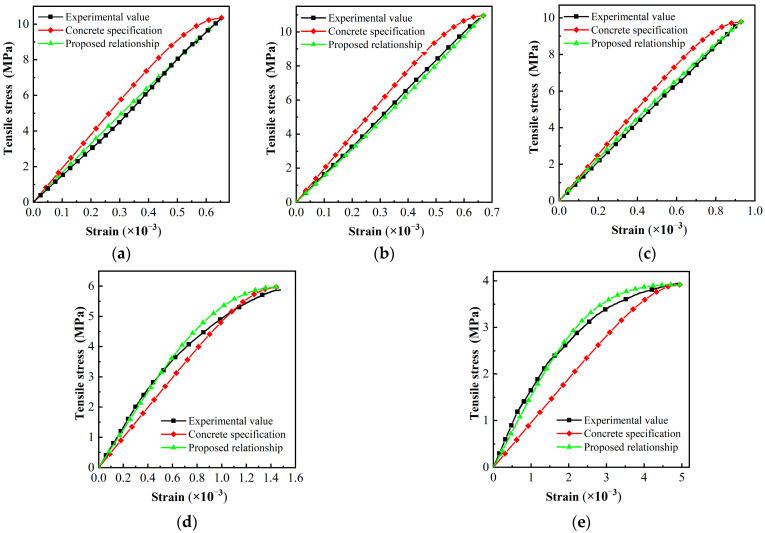
Comparison of tensile constitutive models. (**a**) −10 °C, (**b**) 0 °C, (**c**) 15 °C, (**d**) 40 °C, (**e**) 60 °C.

**Table 1 materials-16-02653-t001:** Aggregate Grading.

Sieve Pore Diameter (mm)		9.5	4.75	2.36	0.6	0.3	0.15	0.075
Passing rate (%)	Gradation upper limit	100	88	75	50	22	10	6
	Gradation lower limit	95	60	41	15	6	3	0
	Designed gradation	100	65	49	28	12	6	3

**Table 2 materials-16-02653-t002:** Mix proportion of modified polyurethane concrete.

Component	Coarse Aggregate	Fine Aggregate	Modified Polyurethane Binder	Curing Agent
Mass fraction (%)	30.2	54.4	15.2	0.2
Fineness modulus	3.4	2.5	/	/
Apparent density (kg/m^3^)	2600	2580	1005	/

**Table 3 materials-16-02653-t003:** Comparison of MPUC contact stretching schemes.

Number	Test Method	Specimen Diagram (mm)	Thickness *w*/Diameter *D* (mm)	Specimen Failure	Damage Feature
SJ-1	S1 + Type I	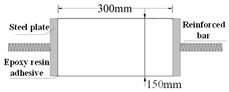	*D* = 150	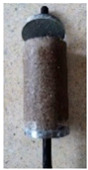	Undamaged specimen, adhesive layer debonding
SJ-2	S2 + Type II	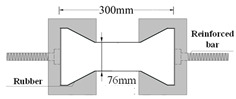	*w* = 75	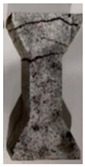	Cracks mostly occur at the loading end, and the stress concentration is obvious.
SJ-3	S3 + Type II	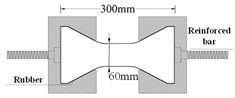	*w* = 75	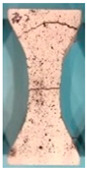	Main crack is in the gauge section, yet an obvious crack appears at the load end.
SJ-4	S3 + Type III	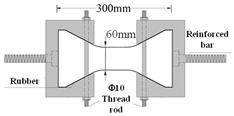	*w* = 75	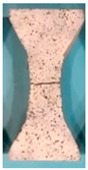	Ideal tensile fracture effect

**Table 4 materials-16-02653-t004:** Results of MPUC uniaxial tensile experiment.

Temperature (°C)	Force (kN)	Peak Stress (MPa)	Peak Strain-Strain Gauge (10^−3^)	Peak Strain-DIC (10^−3^)	Relative Error (%)	Elastic Modulus (GPa)	Fracture Energy Density (N·mm^−2^)
−10 °C	46.26	10.28	0.681	0.618	9.25	16.86	3.304
0 °C	48.60	10.80	0.690	0.623	9.71	16.16	3.929
15 °C	44.06	9.79	0.958	0.899	6.16	11.93	4.247
40 °C	26.42	5.87	1.432	1.524	6.42	6.23	5.561
60 °C	17.78	3.95	4.967	4.873	1.89	1.21	12.843

**Table 5 materials-16-02653-t005:** The fitting values of *α* and *β* under five groups of temperatures.

Temperature (°C)	−10	0	15	40	60
*β*	0.93	1.05	0.94	1.58	2.39

Notes: The correlation coefficients are above 0.95.

## Data Availability

Not applicable.

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
