# Peer review of "Study on Temperature-Dependent Uniaxial Tensile Tests and Constitutive Relationship of Modified Polyurethane Concrete"

_materials, 2023, doi:10.3390/ma16072653_

Round 1

Reviewer 1 Report

In this paper, the tensile properties of the modified polyurethane concrete (MPUC) in the diurnal temperature range of steel deck were studied via conducting tensile tests on three kinds of specimens. The topic is interesting. However, there are several concerns about this manuscript which needs to be addressed before its acceptance.

1)     The abstract is too lengthy. Concise rephrasing of the abstract is recommended. In its current form, scattered and illogical structure of the information is presented in the abstract.

2)     Keywords need to be modified and arrange in an alphabetic sequence. Please use words not combinations of words or phrases.

3)     It is recommended to avoid compounding of references.

4)     Introduction section lack critical review of the previous literature.

5)     Motivation of the current study in terms of the shortcoming in the literature is missing.

6)     Organization of the manuscript at the end of introduction section is recommended.

7)     The gradation curves of different aggregates shown in Fig. 2, needs to be discussed quantitively.

8)     A brief reasoning/justification should be provided for the selection of ratios of MPUC (Table 1) used in the current study.

9)     How does the length of the specimen effect the test results?

10) What was the effect of loading speed on the accuracy and validity of the test results?

11) It is not clear how does the strain concentration of specimens as shown in Fig. 7 were obtained. More explanation/clarity is required in this regard.

12) What type of resistive strain gauges were used in the experiment? What was the accuracy and sensitivity?

13) Procedure of identifying the black dots on the specimen surface should be added to the manuscript.

14) The key challenges should be identified and recommendations for work should be provided in the conclusion.

15) The linguistic quality of the paper is also weak.

Reviewer 2 Report

In my opinion, the scientific article submitted for review is well written. The authors conducted an excellent review of the literature, planned the experiment, described the results obtained, performed a whole series of analyzes and evaluation of the results obtained, and introduced elements of statistical analysis into the paper.

The manuscript is good, but I have a few comments that should be clarified by the authors - some of them decided that I did not give the final grade as "minor revision", but "major revision".

The manuscript should include nomenclature - a list of symbols and all markings - in order not to disturb the layout of the MDPI Publisher, it may be placed at the end of the paper - some publishers do so.

In the paper, in relation to the broadcast material, please do not use the word "sample" - the preferred word "specimen" in accordance with generally accepted standards.

In the literature review, there are editing errors in several places - spaces, commas are missing, especially when quoting other literature items - please check the manuscript in this respect.

I recommend adding a technical drawing for specimen 1 to the paper - please complete figure 4 in the manuscript.

Any dimensions in the figures in table 2 should be moved outside the specimen outline.

I would supplement table 3 with the values of force and displacement to achieve "peak stress" - necessarily.

The manuscript lacks information about the repeatability of research - this is basically my only complaint against the authors of the paper. We have tests of a very brittle material here, but we do not know its repeatability. The authors should, for each of the test temperatures, carry out at least three tests, show the obtained graphs of force versus displacement and stress versus strain. Determine the results we are interested in, draw the lower envelope of the obtained curves on the obtained graphs. The determined parameters should be subjected to a full statistical analysis - minimum, maximum, average, median, scatter, measurement uncertainty. Then we would have full information about the study. Now we basically know that something has been done and that's it. Fragile materials have the advantage that the dispersion of results can be significant. The authors say nothing about it. Please be sure to clarify my doubts, it is best to supplement the paper with the elements I have listed above.

The analysis of the results is interestingly developed, but it is not without flaws. Let us write the coefficient of determination determining the fit of the equation to the experimental data as "R2" - the authors use "R2" - please correct it.

In the case of the analysis of Young's modulus, the authors made a mistake in the unit - see Figure 16. Young's modulus is expressed in "GPa" The authors in Figure 16 gave the unit "MPa" - so three orders less - I assume this is a mistake, not a mistake substantive.

In addition, the manuscript is interesting and worth recommending.

Please correct the paper and resubmit for review.

Round 2

Reviewer 1 Report

Authors have significantly improve the manuscript. I would like to recommend the manuscript for publication in its current form.  

Reviewer 2 Report

The authors included all my suggestions in the revised version of the paper. I recommend the manuscript for publication.